# Experimental Research and Numerical Analysis of the Elastic Properties of Paper Cell Cores before and after Impregnation

**DOI:** 10.3390/ma13092058

**Published:** 2020-04-29

**Authors:** Michał Słonina, Dorota Dziurka, Jerzy Smardzewski

**Affiliations:** 1Department of Furniture Design, Faculty of Wood Technology, Poznan University of Life Sciences, Wojska Polskiego 28, 60-637 Poznan, Poland; michal.slonina@up.poznan.pl; 2Department of Wood-Based Materials, Faculty of Wood Technology, Poznan University of Life Sciences, Wojska Polskiego 28, 60-637 Poznan, Poland; dorota.dziurka@up.poznan.pl

**Keywords:** honeycomb, cell geometry, starch, sodium silicate, epoxy resin, DIC, FEM

## Abstract

The research hypothesis states that the impregnation of the honeycomb paper core of lightweight sandwich panels with modified starch, sodium silicate and epoxy resin (LiquidWood^®^) resin has a significant effect on its elastic properties. In this study, a recycled paper was used in three thicknesses, seven types of cell shapes, including two after numerical optimization and three types of impregnating agents. The method of digital image analysis determined the elastic constants of manufactured paper cores, which were subjected to axial compression in two directions. Based on the experimental results, elastic constants of the cores were calculated and compared with the results of numerical calculations. It has been shown that each of the impregnating solutions used improved the stiffness of the paper core. The best results were obtained for LiquidWood^®^ epoxy resin and modified starch. An important parameter of cell geometry affecting their rigidity is the angle of the cell wall φ, as well as the arrangement of the common cell wall in relation to the direction of load. The numerical models developed were positively verified.

## 1. Introduction

Lightweight sandwich panels with a paper core are successfully used in many industries. Dominates especially in the aerospace industry [1,2], shipbuilding [3,4,5], automotive [6], buildings construction and in the furniture industry [7,8,9,10]. The cores of such panels most often have a honeycomb structure providing the panels with satisfactory mechanical properties due to the excellent ratio of relative density to their stiffness and strength [11,12,13,14]. For practical reasons, elastic properties and relative density of polygonal core cells of sandwich panels were also modeled [15,16]. In many of these studies, the external dimensions of cells were recorded by equations omitting the cell wall thickness, treating it as an integral, but not autonomous, component of the wall length. This approach significantly limits the unambiguous analysis of the effect of wall thickness on the relative density of the cell. There are also known structural solutions of sandwich panels made of light metals or polymers [16,17,18], as well as modified Nomex papers or aramid fibers (Kevlar) [11]. The properties of these composites using various facing materials have been tested, among others, by [19,20,21].

Among the materials of biological origin, the paper is most often used for the production of cell cores. However, forests cover as main resources of natural fibers are shrinking. This enforces the necessity to search for new, more rational ways of using them. Lightweight cellular boards have been implemented in the furniture industry directly from door manufacturing factories [5,22]. They successfully replace most common wood fiber materials such as: particleboard (PB), plywood (PW), medium density fiberboard (MDF), hard fiberboard (HDF), oriented strand board (OSB), the density of which ranges from 550 kg/m^3^ to 950 kg/m^3^ [23,24]. The honeycomb furniture board consists of two rigid and thin faces, usually made of HDF boards, bonded with adhesive to a paper core [25]. However, there are strong limitations on the widespread use of these boards for furniture design. Barboutis and Vassiliou [26], Smardzewski et al. [27], Smardzewski and Jasińska [28] note that horizontal furniture construction elements like shelves can only be designed from honeycomb boards not less than 25 mm thick.

Research to date has led to improvements in the rigidity of cell plates with a hexagonal core by changing the size and shape of cells [29,30,31]. The thickness and type of face material were also investigated [32,33]. However, these solutions did not provide the cellular board with strength and rigidity corresponding to the properties of P2 particleboard furniture of comparable thickness. Sam-Brew et al. [33] also identified factors affecting the quality of boards with hexagonal cells made of paper. These were: Cell size, core density, and cell orientation relative to the manufacturing direction of the board. It has been proven that sandwich board with a core in which common cell walls are oriented in the direction of web formation have the highest values of shear modulus and linear modulus. Bitzer [11], Smardzewski [30] proved that the core built of irregular hexagonal cells placed between two HDF boards well balances the stress generated in the faces. Majewski and Smardzewski [34] showed that increasing the inclination angle of cell walls significantly improves the strength and stiffness of cell plates.

Nowadays, the use of advanced numerical analysis to optimize the mechanical properties of a paper core composite is a common approach. However, there are a few comparisons of analytical and numerical calculations, experimentally verified for elastic properties of cells and cores made of wood materials. In most cases, these studies focus on experimental work and the description of the physic-mechanical properties of corrugated paper [35,36,37,38]. There are also a few attempts to replace the real models of paper cellular cores with simplified models by their homogenization [39,40,41,42,43,44]. The authors of these works using the theory of laminated boards, homogenize the elastic properties of sandwich boards by replacing the cell structure to a material with equivalent thickness and orthotropic properties. For the practice of modeling with the finite element method, this is of fundamental importance, as it significantly reduces the calculation time. Librescu and Hause [45], Aboura et al. [35], Biancolini [38] report that the stiffness of sandwich panels, with a cellular core, is most often modeled by adjusting the thickness of the inner layer. Usually, increasing the thickness of the core leads to an increase in the rigidity of the entire structure, at the expense of a slight increase in the weight of the finished composite. In addition, through appropriate modifications to the core properties, an increase in their sound absorption, fire resistance or resistance to changing temperature conditions and humidity can be achieved [46]. Pohl [47] in his work showed that hexagonal paper filling during the change of climatic conditions might lose over 70% of the original compressive strength. Paper absorbing water behaves similarly to wood [48,49]. One of the ways to protect the core against the damaging effects of moisture is its impregnation. Attempts at impregnating paper intended for cellular cores were made by [3,47,50,51], among others. These authors used the following hydrophobic agents: Liquid phase elastomers, polydimethylsiloxanes, commonly available silane based wood preservatives, synthetic resins (phenolic, urea, epoxy), polyvinyl acetate, nitrocellulose varnishes, polystyrene solutions with ethyl acetate, linseed oil, aluminum acetate. The material’s sensitivity to moisture was studied using compression tests. It was found that the compressive strength of the paper honeycomb in humid environments can be as low as 25% of its strength in dry environments. However, strength tests showed that some substances (polydimethylsiloxanes, nitrocellulose varnishes, solutions of polystyrene with ethyl acetate, linseed oil, aluminum acetate) have a negative effect on the mechanical properties of the paper core. In the work [47] an environmentally friendly, impregnation liquid was developed, which succeeds in preserving the structural strength of the paper honeycomb under high-moisture conditions and also renders the material noncombustible. Absorption of water vapor from the air has a significant impact on the strength of wood materials, including light sandwich panels. As noted in [46,52] there are a limited number of publications dealing with the strength of cell plates in changing climatic conditions. In the conducted study, the authors proved that furniture joints made of honeycomb boards in the climatic conditions of the production hall in Poland, may lose up to 25% of their original rigidity and 40% strength in climatic conditions of the tropical zone. Given the constant expansion for new markets of Polish furniture, such a significant problem should not be neglected.

The above discussion reviles that new lightweight sandwich furniture boards made of wood-based materials are a desirable composite not only in the furniture industry. However, there is a lack of basic characteristics that would allow designers to design and calculate optimal, lightweight constructions made of these wood-based composites. The number of publications, dealing with the behavior of such composites in a tropical climate, is insufficient. The basic standard EN-318 [53] provides a unified method for assessing the impact of changes in air humidity on the properties of wood materials at a constant temperature of 20 °C, but it does not take into account the conditions that occur in tropical zone countries. A lack of this knowledge usually leads to the redesign of components of layered composites and the use of materials not adapted to carry the designed loads. Such design method is economically unjustified and contradicts the rational economy of shrinking resources of natural materials. There are also no guidelines as to the possibility of limiting the adverse effects of increased temperature and humidity on the strength and stiffness of lightweight sandwich panels with a paper core. Considering the above, the following research hypotheses were formulated. The impregnation of the paper core cell walls with selected agents (modified starch, sodium silicate, liquid wood) has a significant effect on the elastic constants of the core.

The main purpose of the research was to determine how impregnation of paper with selected agents (modified starch, sodium silicate, liquid wood), as well as the shape and dimensions of the hexagonal cells obtained from this paper, affect Poisson’s ratio and modules of linear elasticity of cores. The cognitive goal of the work was to verify the quality of numerical models by comparing the results of computer calculations with the results of experimental research.

## 2. Materials and Methods

### 2.1. Choice of Papers and Impregnating

Testliner 2 a paper produced by HM Technology (Brzozowo, Poland) was used for the tests, in three thicknesses: 0.1 mm, 0.15 mm, 0.25 mm and grammage respectively: 85 g/m^2^, 123 g/m^2^ and 134 g/m^2^. Paper with a grammage 120 g/m^2^ and thickness 0.15 ± 0.02 mm is the basic raw material used by the manufacturer of paper fillings, Axxor (Zbąszynek, Poland), a leader in the production of hexagonal paper honeycombs. Also, the literature describes the use of paper of this thickness [33]. The other two paper thicknesses (0.1 and 0.25) were selected by means of numerical optimization using the Monte-Carlo method [54].

As paper impregnation agents it was decided to use: acetylated starch (ST) (patent number P.430486), an aqueous solution of sodium silicate (SS) (sodium silicate $Na_{6}Si_{2}O_{7}$) and epoxy resin LiquidWood^®^ (LW). Modified starch is a polysaccharide that has undergone chemical processes to highlight the desired physico-mechanical properties. The aqueous solution of sodium silicate, manufactured by Dragon Poli (Krakow, Poland), is a result of the chemical reaction of molten sodium hydroxide or sodium carbonate with silica, LiquidWood^®^ is a two-component epoxy resin with zero VOC emission which confirms the Greenguard^®^ certificate. Modified starch and an aqueous solution of sodium silicate, at doses below 1.59 mg/kg, are considered to be non-toxic. Polysaccharides are used in the paper industry to improve the mechanical strength of paper and increase dimensional stability in changing climatic conditions. Acetylated starch has not been used so far. An aqueous solution of sodium silicate is most often used for surface insulation of concrete against moisture. It also creates refractory coatings. LiquidWood^®^ epoxy resin is an innovative product that is particularly used to reconstruct fragments of wooden structures decrepit or damaged by living organisms. This epoxy resin penetrates and hardens deteriorated wood, restoring its strength and function. The purpose of the research was to select impregnating agents that will be primarily neutral to the natural environment and will provide opportunities for recycling. Notwithstanding the improvement in the mechanical properties of the paper saturated with them.

Sheets of raw paper were surface impregnated on both sides with the help of selected substances, applying them using a paint roller in the amounts given in Table 1. Then, the paper was air-conditioned in the conditions of the production hall, in which the air temperature was 25 °C and the relative humidity of air 45%.

### 2.2. Method for Determining Ultimate Strength of Papers

Strips of paper 15 mm × 220 mm were prepared to determine the elastic properties. For each type of unimpregnated and impregnated paper, taking into account its orthotropy (machine direction Y and transverse to machine X), in total, 10 samples were used, which came to 320 pieces. The elastic properties of paper, including linear elasticity modules $E_{x}$ and $E_{y}$, as well as Poisson’s coefficients $\upsilon_{xy}$, $\upsilon_{yx}$ for machine direction (Y) and perpendicular to the machine direction (X), were determined in accordance with PN-EN ISO 1924-2 [55]. Prepared samples were conditioned before testing in the conditions of the production hall, in which the average annual temperature of 25 °C and the average relative humidity of 45% was recorded. Uniaxial tensile tests were carried out on a universal Zwick 1445 testing machine (Zwick GmbH, Ulm, Germany) using an optical Dantec Dynamics extensometer (Dantec Dynamics A/S, Denmark).

### 2.3. Choice of Cell Geometry

Core samples with hexagonal cells were formed from the impregnated paper. The shape of the A-E cells was chosen by analytical method. Whereas, F-G is based on the numerical optimization algorithm. Both methods were presented in detail in [54]. The cell dimensions obtained in this way are summarized in Table 2, and to better distinguish differences in cell geometry, their shapes along with symbols are illustrated in Figure 1.

As can be seen from Table 2, cells (A–E) with constant relative density and paper thickness were selected for testing. The paper geometry and thickness of the last two cells have been subordinated to maximization of linear elasticity modules for F and minimization of paper thickness for G.

### 2.4. Method of Manufacturing Core Cells

Samples consisting of 3 to 6 cells evenly spared along the perpendicular edges of the sample were prepared for testing (Figure 2). Samples were prepared and conditioned in the environment of the production hall where the average annual air temperature was 25 °C, and the relative humidity was 45%.

Properly prepared sheets of unsaturated and saturated paper with selected impregnates were subjected to burnishing (Figure 2A) and cutting into strips with a width corresponding to the height of the core. The operations were carried out using the Kongsberg-X plotter (Kongsberg, Norway). Before applying the adhesive to the surface of the burnished paper, appropriate templates (Figure 2B) were prepared to allow for precise application of the adhesive. Figure 2C shows a template with a paper sheet inside. The glue was applied to the packet prepared in this way using glue rollers. Figure 2D shows a sheet of paper with adhesive applied. Subsequently, the sheets of paper were assembled into sets, 7 MPa was applied and seasoned for a period of 7 days in the climatic conditions of the production hall. The final stage was cutting the bonded sheets of paper into 17 ± 0.1 mm wide strips (Figure 2E), from which a core is then formed. PVAc Woodmax FF12.47 class D2 adhesive from Sythos Adhesives (Oświęcim, Poland) was used to bond the paper strips into the core. It is a white glue, which is an emulsion of polyvinyl acetate in water and is biodegradable. PVAc is non-toxic to humans. However, it does release toxic fumes if burned. The adhesive was applied with adhesive rolls at the amount of 40g/m^2^ (Figure 2E). The glueing process was carried out in the Orma Macchine NPC/DIGIT 6/90 25 × 13 plate press (Bergamo, Italy) for 10 min. at 70 °C. Stretch templates to achieve the desired core cell shape (Figure 2F) were modeled in 3D CAD SolidWorks 2016, Dassault Systèmes SolidWorks Corp. (Waltham, MA, United States), and then printed using Nylon-12 ™ filament (Stratasys, MN, USA) on a 3D printer Stratasys Fortus 400mc small (Stratasys, MN, USA) working in FDM (Fused Deposition Modeling) technology. Prepared and glued core strips were manually pulled into a three-dimensional mold (Figure 2F). Then the core shape was fixed at 80 °C for about 240 s. For each cell type, type of paper and impregnation and compression direction X and Y, 5 samples were made, 140 in total.

### 2.5. Method for Determining the Elastic Properties of Cores

Cores made of non-impregnated and impregnated paper (Figure 3a) were subjected to uniaxial compression in the Y and X directions, under climatic conditions corresponding to the production hall (temperature 25 °C, relative air humidity 45%). The tests were performed on a Zwick 1445 testing machine (Zwick GmbH, Ulm, Germany). The test stand consisted of a traverse (A) and support (C) on which the core (D) was placed in a transparent rectangular box protecting the core against buckling (B). The stand was illuminated with two LED lamps with a brightness of 630 lumens each.

Monochrome photos of the core sample before deformation and during loading were taken ten times with an interval of three seconds, using the Olympus OM-D camera (Olympus, Tokyo, Japan) (F). The length standard (E) with an accuracy of 0.01 mm was set on the support. Travers compressed the core at a speed of 10 mm/min over a distance of 5 mm. At the same time, the measuring head of the testing machine registered 30 force measurements with an accuracy of 0.01 N. The measurements of the core’s linear deformations were made by the edge detection method (Figure 3b), analyzing the displacements recorded in the pictures and comparing them with the calliper metric standard (accuracy 0.01 mm). The values of the displacements read were compared with the appropriate force values. Digital image analysis was performed using the National Instruments IMAQ Vision Builder 6.1 software (National Instruments, Austin, TX, USA). Figure 3b illustrates how to measure displacements in the direction of loading.

The corresponding displacements for the load in the X direction were recorded as longitudinal Δ*_x_* and transverse Δ*_y_*. Understanding the value of the force acting on the X direction, the value of the linear modulus of elasticity $E_{x}$ and the Poisson’s ratio $\upsilon_{xy}$ were calculated from the equations,
(1)$$\sigma_{x}=E_{x}\varepsilon_{x}$$
hence,
(2)$$E_{y}=\frac{PS_{y}}{A\Delta_{y}}$$
(3)$$v_{yx}=\frac{\varepsilon_{x}}{\varepsilon_{y}}$$
where: $\sigma_{x}$–normal stress for the X direction, $\varepsilon_{x}$-deformation for the X direction, $\varepsilon_{y}$-deformation for the Y direction, A-force impact surface, P-force, $L_{x}$ and $S_{y}$–core dimensions. To determine the $E_{x}$, $\upsilon_{xy}$ the same procedure was followed using the below formulas:(4)$$E_{x}=\frac{PL_{x}}{A\Delta_{x}}$$
(5)$$v_{xy}=\frac{\varepsilon_{y}}{\varepsilon_{x}}.$$

### 2.6. Numerical Models of Cores

Numerical calculations were aimed to estimate the quality of these models in relation to the results determined in the laboratory. To this end, it was necessary to map appropriate cell shapes based on the geometry of the cores obtained in reality, in particular taking into account their imperfections. Calculations using the finite element method were performed using Abaqus 6.13-1 software (Dassault Systemes Simulia Corp., Waltham, MA, USA) and the resources of the EAGLE computing cluster, Poznań Supercomputing and Networking Center. Hex type mesh and elastic-plastic deformations were used. The number of nodes in the mesh was in the range of 0.3–1.8 million and the average number of finite elements 0.32 million. C3D6 and C3D8R solid elements were used. The core models were assigned elastic properties of paper determined in experimental studies. The models were subjected to axial compression with a displacement of up to 5 mm. The value of this displacement corresponded to the results of laboratory tests. To determine the values of elasticity constants $E_{y}$, $\upsilon_{xy}$ and $E_{x}$, $\upsilon_{yx}$ proceeded in the same way as during experimental research, following appropriate displacements in the direction of the force and transverse to this direction. Then the appropriate values were calculated from the Formulas (2)–(5).

## 3. Research Results and Their Analysis

### 3.1. Influence of Impregnating Agent Concentration on Paper Strength

The expected positive effect of the impregnating agents used was confirmed during a uniaxial tensile test of paper strips. Based on the results presented in Figure 4, Figure 5 and Figure 6, the right concentrations of agents were determined.

Figure 4 shows that the ultimate tensile strength was obtained by starch impregnated paper with 10% and 20% solution concentration. It was, therefore, considered that this difference is insignificant for the strength of the paper, but important from the cost point of view of impregnation. In addition, in relation to the strength of unimpregnated paper, 10% and 20% solutions increase the strength of impregnated paper by 28.9% and 31.1%, respectively. To sum up, a 10% aqueous solution of modified starch was used for further studies.

Based on the results summarized in Figure 5, it was decided that for further testing, the paper was impregnated with a 40% water glass solution with distilled water. However, Figure 6 shows that impregnating paper with LiquidWood resin provides the highest strength compared to other agents and raw paper.

### 3.2. Elastic Properties of Hexagonal Cores

Experimental studies allowed to determine the most important elastic constants of cores made of unimpregnated and impregnated paper. The second and third columns of Table 3 contain the values of Poisson’s ratio and further modules of linear elasticity. This table shows the fundamental regularity that the Poisson’s ratio $v_{xy}$ usually are larger than $v_{yx}$. Another regularity is the higher value of the *E*_x_ module in relation to *E*_y_. This is mainly due to the structure of the cells and the way the samples are loaded along the length of the common cell wall. Common cell walls significantly enhance the rigidity of the hexagonal core cells. For this reason, during compression in the X direction (Figure 7a), the free walls of the cell are bent and then the joint of the cell walls are forced to lose stability, whereas during compression in the Y direction only the free walls of the cell are bent (Figure 7b).

This behavior of cells reviles strong core orthotropic, which will undoubtedly affect the orthotropic properties of the lightweight sandwich board produced from them. As can be seen from further analysis of Table 3, the smallest linear modulus of elasticity $E_{x}$ = 0.0019 (MPa) is characterized by the non-impregnated C-type cell core and the E-type core ($E_{y}$ = 0.0006 MPa). At the same time, this cell has the lowest Poisson’s ratio $\upsilon_{yx}$ = 0.06 and the largest $\upsilon_{xy}$ = 8.72. In the group of cells impregnated with sodium silicate solution, the lowest Poisson’s ratio ($\upsilon_{xy}$ = 0.77) is the C_SS core. The E_LW core impregnated with LiquidWood^®^ epoxy resin distinguishes itself by the largest modulus of linear elasticity ($E_{x}$ = 0.1185 MPa) and the strongest orthotropy. The impregnation with modified starch A_ST causes the highest value to be reached by the module $E_{y}$ = 0.0057 (MPa). In contrast, the largest Poisson’s ratio $\upsilon_{yx}$ = 0.80 was recorded for the C_ST core. From Table 3, it can be generalized that all the impregnation methods used have brought the expected effect. The elastic properties of the cores after impregnation have significantly improved. For cores impregnated with modified starch, $E_{y}$ increased from 88% to 515% and $E_{x}$ increased from 89% to 1209%. Similarly, cores impregnated with sodium silicate solution increased $E_{y}$ values from 38% to 550% and $E_{x}$ values from 14% to 1033%. In the case of LiquidWood epoxy resin, linear elasticity moduli increased from 8% to 500% and from 300%, to 2852%, respectively. Among all impregnation methods, the greatest effect on increasing the linear modulus $E_{y}$ have modified starch, and $E_{x}\;$ LiquidWood^®^ epoxy resin.

Further data analysis shows that among the cores made on the basis of cells after Monte-Carlo (F-G) optimization, the G cell is characterized by ten times lower $E_{x}\;$ modulus of elasticity in relation to the type F structure. The highest value of the elastic modulus $E_{x}$ = 1.28 MPa was obtained for core with F_LW cell impregnated with epoxy resin. At the same time, the same core also has the largest module $E_{y}$ = 0.0108 MPa. The smallest module in the X direction has a G_SS type core impregnated with sodium silicate solution ($E_{x}$ = 0.0272 MPa), as well as the lowest value recorded for the opposite direction $E_{y}$ = 0.0006 MPa. It should also be noted that the highest value of the Poisson’s ratio $\upsilon_{xy}$ = 9.79 was obtained for a non-impregnated F-cell core and the smallest for a non-impregnated G-type core ($\upsilon_{xy}$ = 5.33). For the opposite direction of the test, the coefficient $\upsilon_{yx}$ is in the range from 0.05 to 0.09 for cores with F, G cells. Paper impregnation increases the value of the Poisson coefficient $\upsilon_{yx}$ for F_ST cores to 0.09. However, it was also noticed that the G_ST core, despite the same impregnation method has the lowest Poisson’s ratio value $\upsilon_{yx}$ = 0.05.

### 3.3. Comparison of Experimental and Analytical Results

The dimensions of cores made in reality usually differ from those used in analytical calculations [56,57]. Therefore, for cores made of non-impregnated papers, a digital image analysis method was used to determine the actual geometry of the core cells. Table 4 summarizes the measurement results and dimensions determined analytically in work [54]. As it results from Table 4, the linear dimensions of real cells (R) do not differ significantly from their theoretical (T) equivalents. The differences are in the range from 0.02 mm to 1.48 mm for the free wall length l and from 0.16 mm to 0.8 mm for the common wall length h. The differences in the values of the cell wall angle φ are equal from 1.40° to 15.58°, which has a significant impact on the elastic properties of the paper core. It should also be noted that, in reality, the cell walls of the core, when viewed from top, form a grid of curved lines, not straight lines as assumed in analytical calculations [54]. The magnitude of the differences in the elastic properties of the analyzed core models is illustrated in Figure 8 and Figure 9. From Figure 8 it can be seen that the analytically calculated values of Poisson’s ratios are many times smaller than the values determined experimentally. These differences are particularly visible on the example of a C-type core. Poisson’s ratio $\upsilon_{xy}$ = 0.32 is three times smaller than the experimental value $\upsilon_{xy\left(EXP\right)}$ = 1.03. At the sometime, for $\upsilon_{yx}\;$ this difference is almost 30 times smaller. It can be seen from Figure 9 that the analytically calculated modulus of linear elasticity of the cores are greater than the values determined experimentally. For example, for the same C-type cell, the modulus of linear elasticity $E_{y}$ also deviates greatly from the experimental value, being more than 130 times larger. For the opposite direction of the test (X), the difference in the values of the $E_{x}$ module is approximately 5 times. Equally significant differences occur among cores made of cells after Monte-Carlo optimization. This is shown in Figure 10 and Figure 11.

The main reason for these discrepancies is the assumptions based on which the relevant mathematical models were built [54]. These assumptions idealized the shape and dimensions of the cell walls, as well as the shape and dimensions of the cells themselves. They did not take into account geometric imperfections and the resulting weakening of the stiffness of structures subjected to axial compression. For this reason, later in the work, it was decided that the results of experimental studies would be compared with the results of numerical calculations for all types of cells made of unimpregnated and impregnated papers. Achieving compliance of these results confirms the possibility of applying experimental data and numerical models for further research leading, among others, to a homogenization of honeycomb sandwich structures.

### 3.4. Comparison of Experimental Results and FEM Numerical Calculations

First, the compatibility of numerical calculations results with experimental tests of A, B, C, D and E cores was analyzed. Figure 12 summarizes Poisson’s ratios based on the experiment and numerical analysis. It shows that for $\upsilon_{xy}\;$ numerical values usually dominate the laboratory results, while for $\upsilon_{xy}\;$ the opposite is true. Statistically, the average difference between these quantities is 10%, and 8%, respectively. Of course, there are exceptions. A noticeable difference was revealed for the Poisson’s ratio $\upsilon_{xy}\;$ core not impregnated with an E-type cell. The experimental value was 41% higher than the numerically calculated value. The reason for this difference was the large slenderness of the cell, expressed by the ratio $L_{x}/S_{y}=3.61$, and the resulting larger geometrical imperfections. The same cell after impregnation did not show significant changes in geometry in relation to the geometry of the FEM model. Otherwise, for the core with cell A, impregnated (LW), experimentally determined Poisson’s ratio $\upsilon_{xy}$ = 3.95, and the numerically calculated value is 32% higher. However, it should be considered that the results of numerical calculations correspond well to laboratory values, in taking into account the difficulty in modeling the geometry of cells with geometric imperfections, while maintaining maximum diligence in mapping these inaccuracies.

Figure 13 compares the values of linear elasticity modules. The average difference between the obtained numerical and experimental values was equal to 9%. As can be seen from this figure, most of the results from the FEM analysis are smaller than the results of the experiment. The largest difference was read for an impregnated (LW) core with a type D cell where $E_{y}$ = 0.005 and is nine times smaller than determined in the experiment. For the same core D impregnated (ST), the module $E_{x}$ = 0.009, calculated numerically, was half the value in relation to the experimental value ($E_{x}$ = 0.018). And for the D-type core, the calculated module $E_{y}$ was almost two and a half times larger than the value determined in the laboratory. To sum up, despite small deviations of the obtained calculation results, also, in this case, they can be considered satisfactory and close to the results of experiments.

Figure 14 summarizes the results from the experiment and finite element analysis for F and G cells which shape was determined by numerical optimization [54]. It shows that the Poisson’s ratio $\upsilon_{yx}$ has a very small value in the range of 0.04 to 0.09. With the exception of (impregnated F-core (LW)), the results from numerical analysis were on average smaller by about 20% compared to the results of experimental studies. However, for $\upsilon_{xy}$ most of the results were higher by about 5%. For F and G cores, the corresponding differences were not greater than 30%.

Based on the data presented in Figure 15, it follows that the numerically calculated modulus of linear elasticity $E_{y}$ was characterized by a small value in the range from 0.0003 to 0.0108 MPa. For such small values, the difference from the experimental results was equal to 34% on average. All experimental measurements had higher values in relation to the simulation results. In the extreme case, for the F-type core the difference was equal to nearly 70%. In the second direction of orthotropy, the values of the linear modulus of elasticity $E_{x}\;$ ranged from 0.0128 to 1.28 MPa. The results of numerical analysis for F-type cores were on average smaller by about 23%. In contrast, G-cell-based cores were distinguished by approximately 5% greater results compared to the experiment. The exception here is the core G, whose modulus of elasticity $E_{x}$ read during the experimental study was almost three times higher than during simulation. It should be recognized that the results of numerical calculations correspond well to laboratory values.

It is difficult to compare the obtained results with the other research. No papers were found describing the impact of the paper core cell impregnation method on its elastic properties. In work [58] dynamic testing was conducted on phenolic glass fiber reinforced plastic (GFRP) and phenolic-impregnated aramid paper honeycomb specimens in order to analyze the effect of loading rate on the mechanical behavior. The strain rate effect for the composite material is remarkable and leads to increasing Young’s modulus, tensile strength, tensile failure strain, shear modulus and shear strength. Zinno et al. [59] explained that E-glass-phenolic composite shows a considerable reduction in ultimate strain and stress for the hot-wet and acid conditionings, but the Nomex honeycomb core shows a reduction in ultimate shear strength of about 20% for hygrothermal, freeze-thaw and acid conditionings. These works revealed better properties of impregnated cores. However, it is difficult to transfer the results obtained directly to the issue discussed in this paper.

## 4. Statistical Analysis

Statistical analysis was performed to determine the effect of the impregnation method on the elastic properties of the modeled core cells. The results of one-way ANOVA variance analysis are summarized in Table 5, along with post-hoc contrast analysis. It was carried out within the largest population of cores made of paper 0.15 mm thick. All calculations were carried out using the Statistica 13.1 program (StatSoft Polska Sp.z o.o., Krakow, Poland). The study aimed to check which of the independent variables, cell geometry (Cell type), impregnating substance (Imp. Sub.) and orthotropy direction (Dir.), have a significant impact on the modulus of elasticity of the core. Statistically significant results were assumed for *p* < 0.05. Table 5 presents data for testing the null hypothesis H_0_ -no influence of factors on the linear elastic modulus of the core and alternative hypothesis H_1_-independent variables affect the modulus of linear elasticity of the core. It should be noted from this table that the entire model (Intercept) of variance analysis is significant and the value of Fisher’s statistics *f* = 234.4 indicates a small variance within the group in relation to the inter-group. This represents positive feedback regarding the quality of the collected experimental results. The value of the modulus of linear elasticity is significantly influenced by the type of cell (Cell type), the method of impregnation (Imp. Sub.) and the direction of orthotropy (Dir.). The combined effect of the cell type and the impregnation method (Cell type*Imp. Sub) proved to be irrelevant. Interpreting the value of *p* = 0.0664, it can be said that the result in 93.4% supports the H_1_ hypothesis, however the significance threshold was set at least to 95%.

As already mentioned, both row and column effects of the model are important (Intercept). This means that the variables: Cell type, Imp. sub. and Dir., significantly affect the value of the core elastic modulus. In the case of the Dir. variable, which has only two categories $E_{x}$ and $E_{y}$, it is enough to compare the means in subgroups to conclude that $E_{x}$ = 0.0390 MPa is about 1240% greater than $E_{y}$ = 0.0031 MPa.

For the remaining factors, due to the number of categories greater than two, it was necessary to perform a post-hoc test (Figure 16, Figure 17 and Figure 18). For this purpose, the Tukey’s honest significant difference test (HSD) was chosen, which gave the answer to the following question. Which of the categories of factors and how do they affect the linear modulus of elasticity of the core?

After analysis of the statistical data collected in Figure 16, the following conclusions can be drawn. Cells type A, B, D affect the modulus of elasticity of the core in a similar way as a pair of cells type A, E. Cells A, B, D, E are significantly different from C, which has the lowest average modulus of elasticity MoE = 0.006 MPa, and also the lowest variance. B, C, D cells are significantly different from E. The E-cell geometry is distinguished by the largest mean modulus of elasticity MoE = 0.032 MPa and the largest variance. The average modulus of elasticity of an E-type cell is about 8% greater than the reference A-type cell.

For the factor Imp. Sub. Impregnation, with modified starch and sodium silicate solution, has a similar effect on the modulus of elasticity (Figure 17). All impregnation methods have a positive effect on the modulus of elasticity in relation to test 0. This is a significant percentage increase: *SS* = 308%, *ST* = 345% and *LW* = 740%. LiquidWood^®^ epoxy impregnated cores are characterized by the largest average modulus of elasticity *MoE* = 0.041 MPa and the largest variance. The average modulus of elasticity of cores impregnated with modified starch is 12% higher compared to those impregnated with sodium silicate solution. Although the combined effect of the type of cell and the method of impregnation turned out to be statistically insignificant, it is worth analyzing the graph of this pair in relation to the modulus of linear elasticity. Figure 18 confirms the belief that it is the impregnation of paper with LiquidWood^®^ epoxy resin that most effectively increases the modulus of elasticity of the core. The A-type core is particularly susceptible, as itsmodulus of elasticity increased by 375%, and the E-type cells, respectively, 980% relative to the non-impregnated core. Modified starch also affects all cell geometries, but on a much smaller scale. An increase in stiffness of 192%, and 451%, respectively, was observed for type A and E cells. Given that it is a substance of natural origin used on daily basis in the food industry, the result is impressive. When using a sodium silicate solution, the corresponding increments are 16% and 424%. Cores made using orthotropic C-type cells are almost inert to the effects of impregnating agents except LiquidWood^®^ where *MoE* has been increased by 717%.

## 5. Conclusions

The aim of this work was to verify the research hypothesis that the impregnation of the honeycomb paper core of lightweight sandwich panels with selected agents (modified starch, sodium silicate, epoxy resin) has a significant effect on core elastic properties. Therefore, the impregnation of paper, with selected substances, were examined, as well as the shape and dimensions of the hexagonal cells obtained from paper affect Poisson’s ratios and linear elasticity modules of the manufactured cores. The quality of numerical models was also verified by comparing the results of computer calculations with the results of experimental research. Based on the results obtained, it was shown that the single cell geometry has a significant effect on the elastic properties of the hexagonal paper core. A particularly important parameter is the change of the cell wall angle φ, as well as the arrangement of the common cell wall relative to the direction of load. C-type cells exhibit the most favourable isotropic elastic properties, while F-type cells exhibit the most favourable orthotropic elastic properties. Each of the impregnating solutions used improves the rigidity of the hexagonal paper core. The best results were obtained for the LiquidWood^®^ epoxy resin. Modified starch is an excellent agent for the impregnation of lightweight sandwich panel cores, due to the significant improvement of their elasticity constants. In addition, due to the widespread availability and low price, it has great application potential in the furniture industry. Impregnation of A and E cells, with LiquidWood^®^ resin, contributes to the most beneficial increase of their elastic properties. Geometric imperfections are a key factor that influenced the results of analytical calculations. Idealized analytical models show significant differences in comparison with experimental research. However, the numerical models were verified positively. Thus, using the elastic cores defined in the laboratory, you can numerically analyze the properties of new core and lightweight sandwich panels. Optimization of the cell shape by Monte-Carlo method had the intended effect. The modulus of elasticity of the F-type core is significantly greater than any other tested structure, also after impregnation.

## Figures and Tables

**Figure 1 materials-13-02058-f001:**
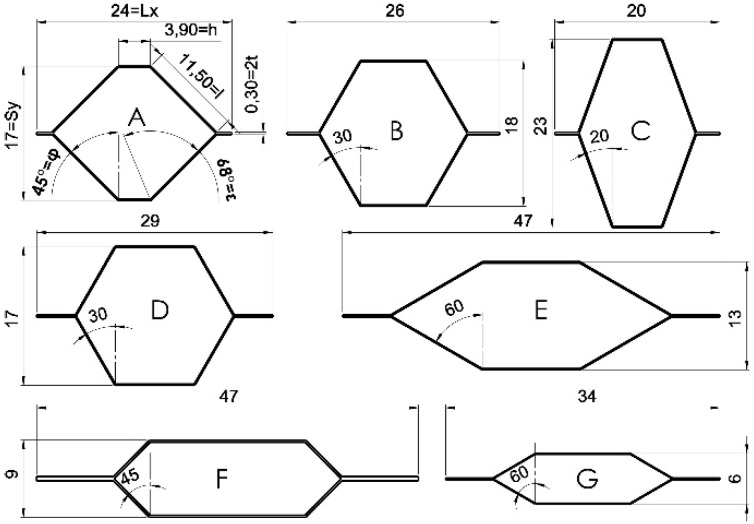
The shape of hexagonal cells.

**Figure 2 materials-13-02058-f002:**
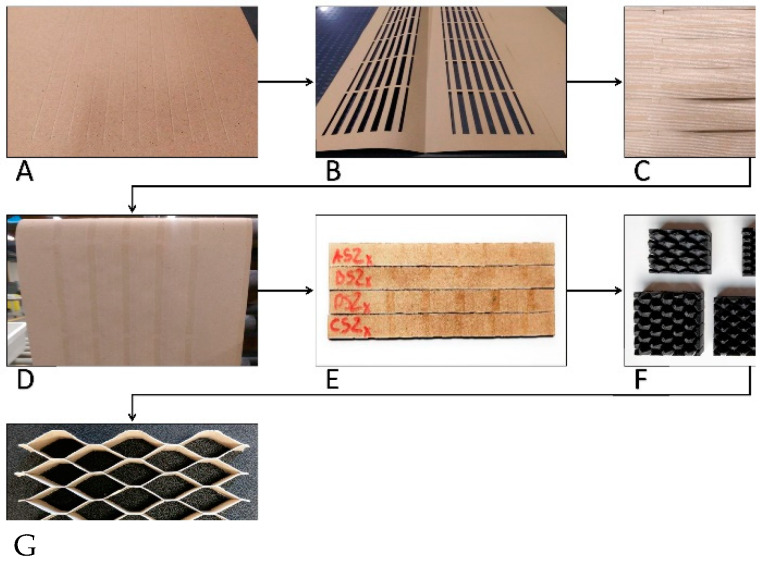
The process of manufacturing the core from sheets of paper: (**A**) burnishing of adhesive lines, (**B**) template for adhesive application, (**C**) folding of template and sheet of paper, (**D**) paper after adhesive application, (**E**) stacked and bonded strips of paper, (**F**) templates for stretching cell, (**G**) core cells after stretching.

**Figure 3 materials-13-02058-f003:**
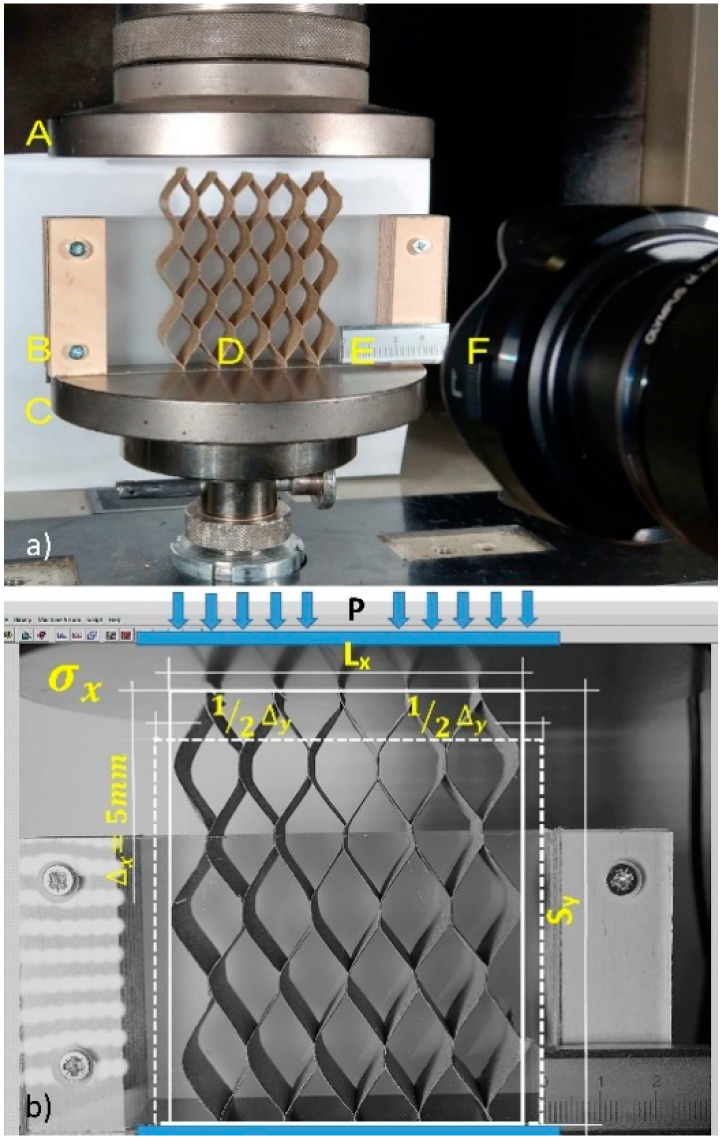
Test stand: (**a**) A—traverse, B—sample cover, C—support, D—sample, E—length standard, F—camera, (**b**) method of measuring core deformations with axial compression of the core, load (P) in the X direction.

**Figure 4 materials-13-02058-f004:**
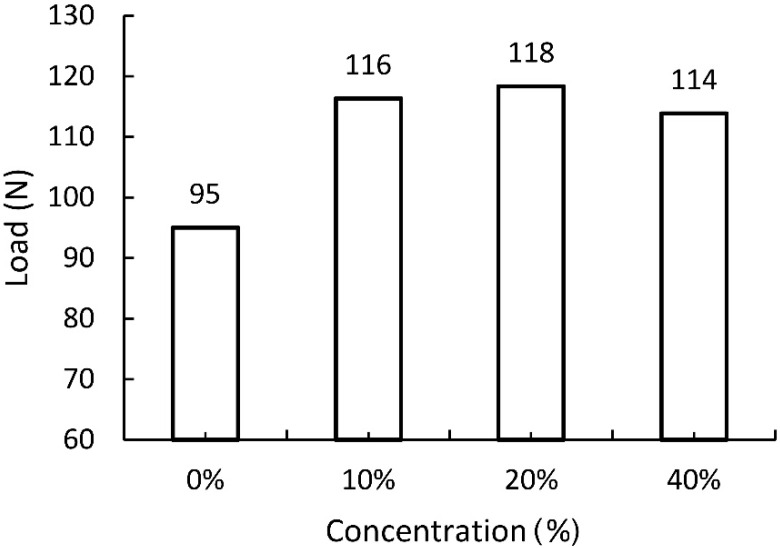
Tensile strength of paper impregnated with modified starch of varying concentration. Unimpregnated paper marked by 0% concentration.

**Figure 5 materials-13-02058-f005:**
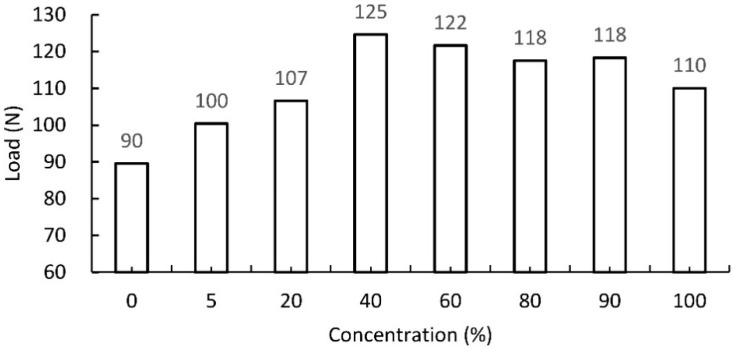
Tensile strength of paper impregnated with an aqueous solution of sodium silicate of varying concentration. Unimpregnated paper marked by 0% concentration.

**Figure 6 materials-13-02058-f006:**
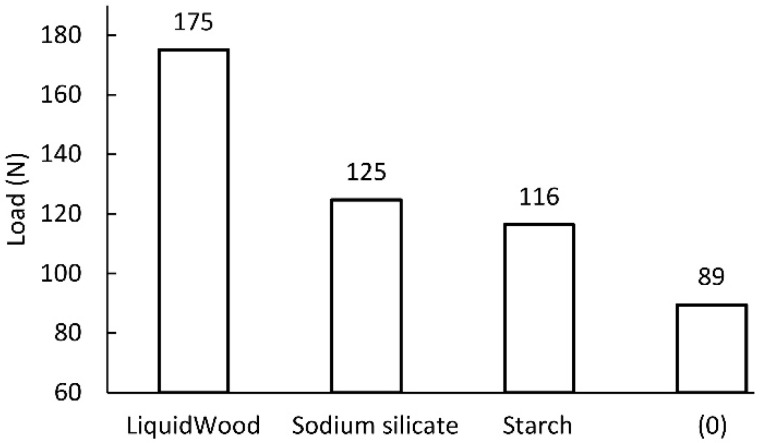
Comparison of tensile strength of impregnated and unimpregnated paper (0).

**Figure 7 materials-13-02058-f007:**
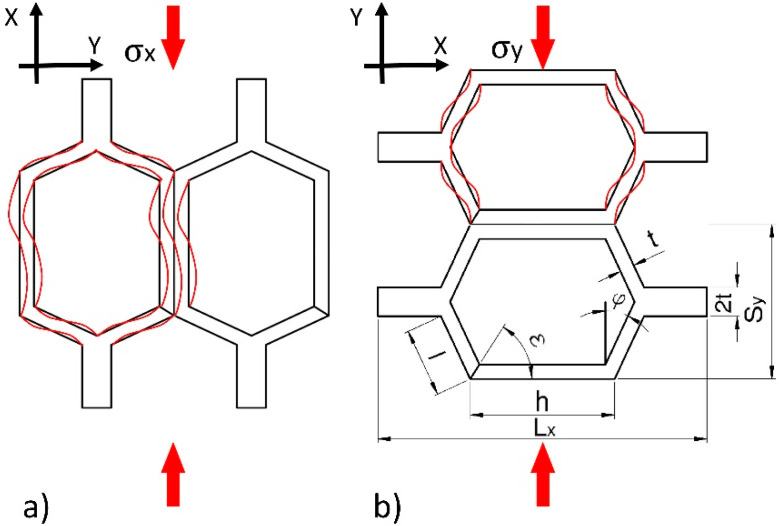
Method of deforming hexagonal cells during axial compression: (**a**) Towards common cell walls, (**b**) perpendicular to the common cell walls.

**Figure 8 materials-13-02058-f008:**
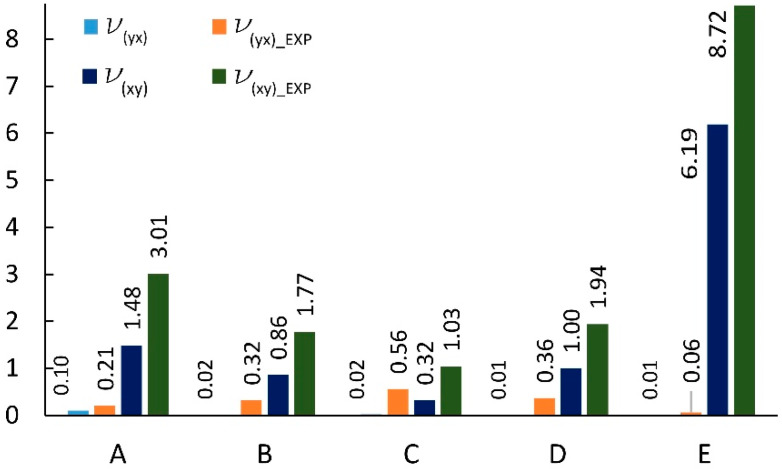
Poisson’s ratio for cells of the type: A, B, C, D, E (EXP-experimental research).

**Figure 9 materials-13-02058-f009:**
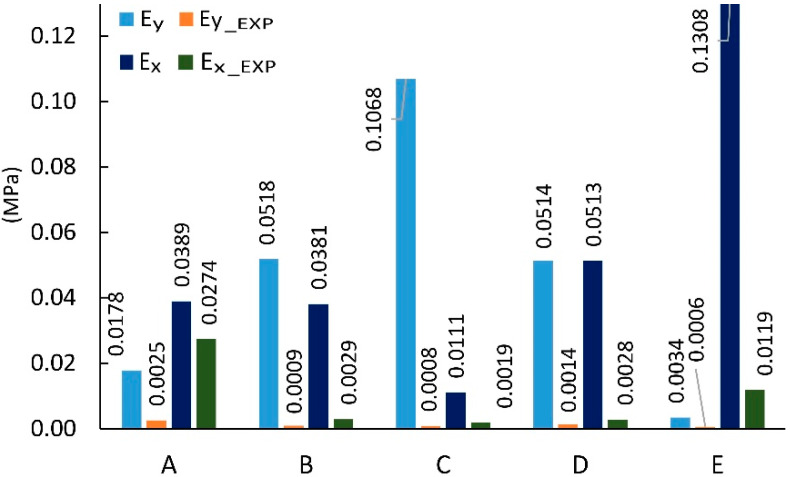
Modulus of linear elasticity for cells of the type: A, B, C, D, E (EXP-experimental research).

**Figure 10 materials-13-02058-f010:**
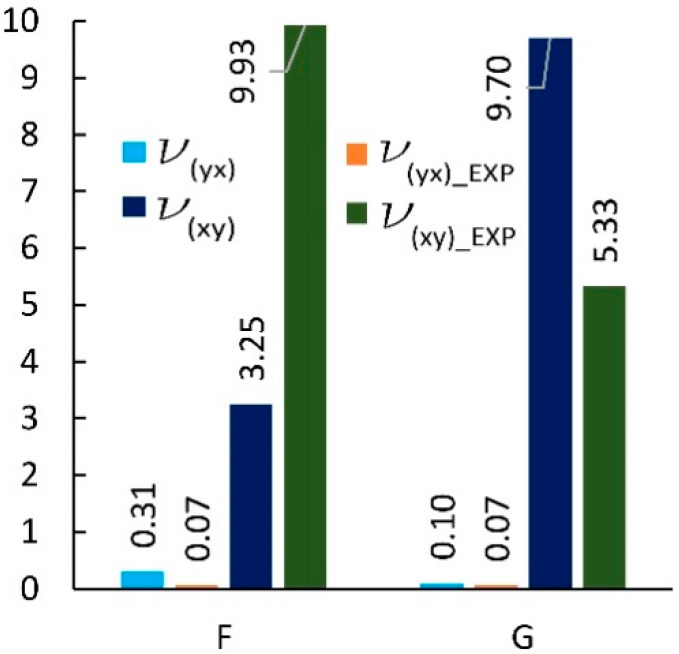
Poisson’s ratio for cells of the type: F, G (EXP-experimental research).

**Figure 11 materials-13-02058-f011:**
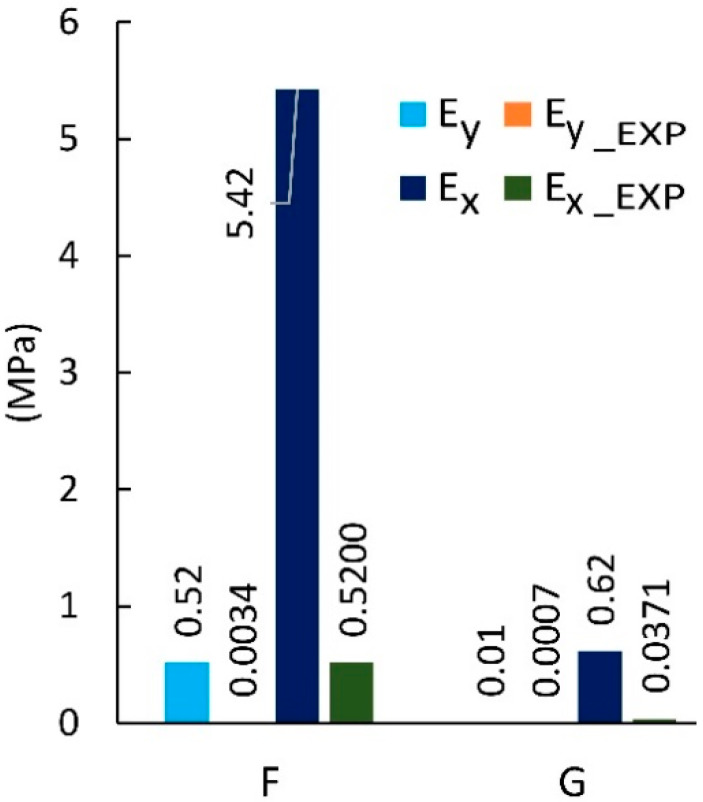
Modulus of linear elasticity for cells of the type: F, G (EXP-experimental research).

**Figure 12 materials-13-02058-f012:**
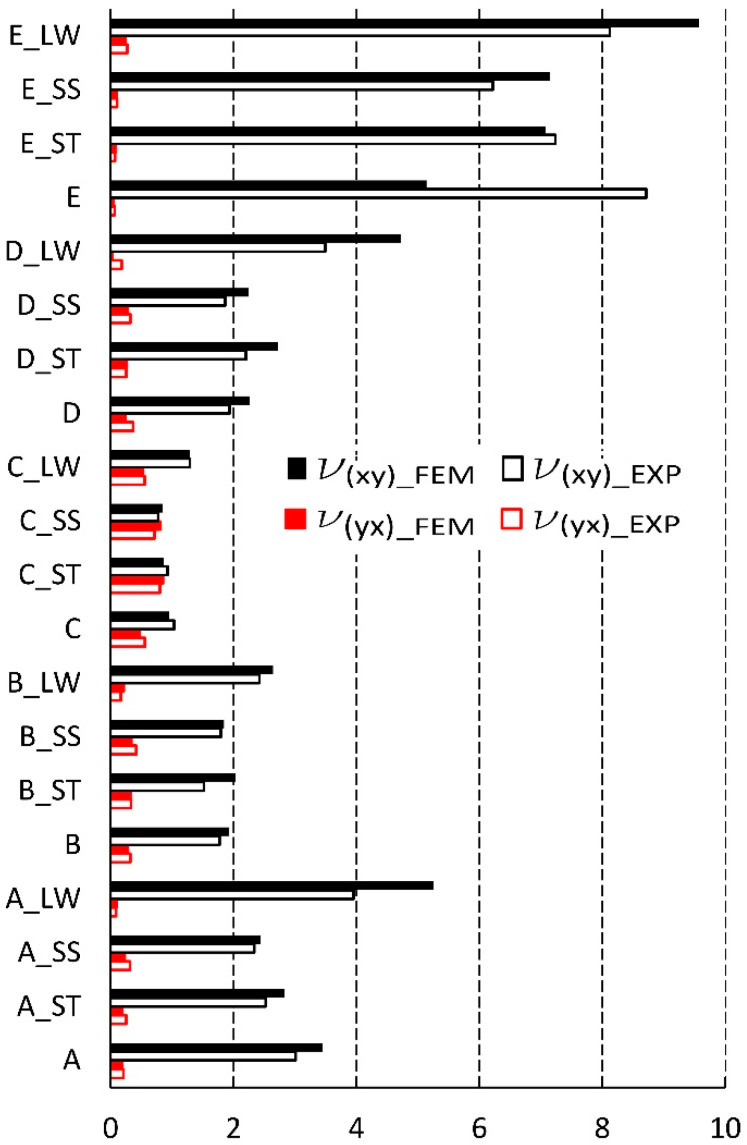
Poisson’s ratio for cell types: A, B, C, D, E (EXP-experimental research, FEM-numerical calculations).

**Figure 13 materials-13-02058-f013:**
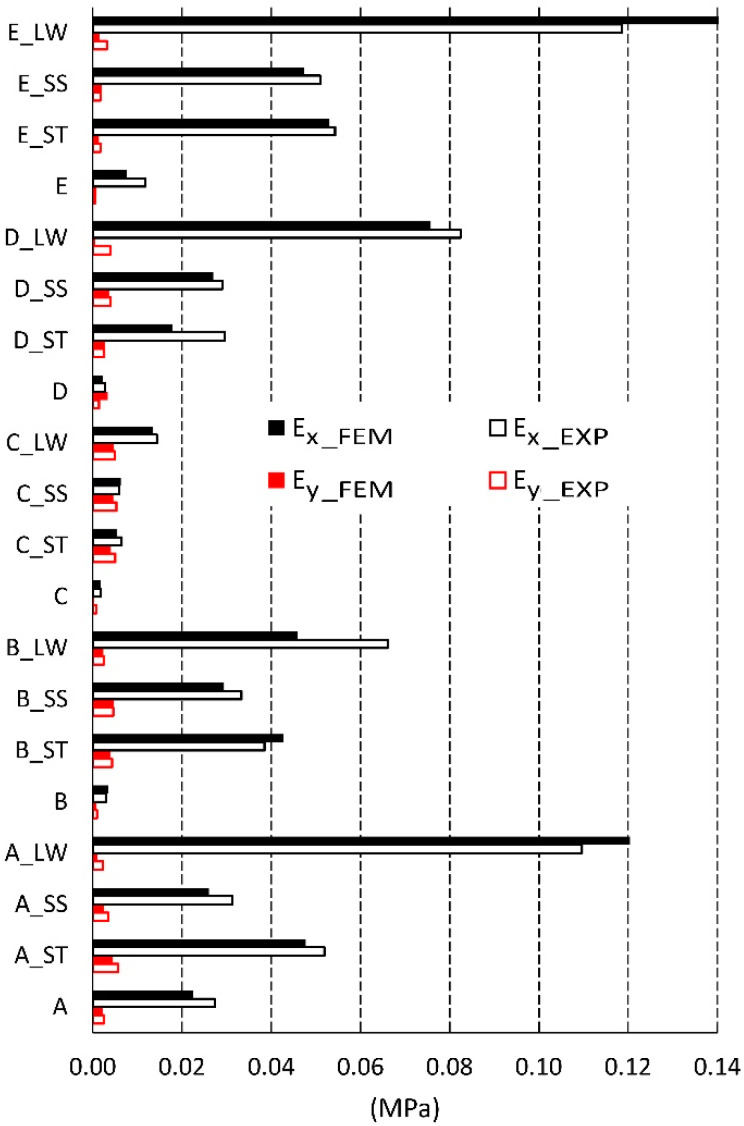
Modulus of linear elasticity for cells of the type: A, B, C, D, E (EXP-experimental research, FEM-numerical calculations).

**Figure 14 materials-13-02058-f014:**
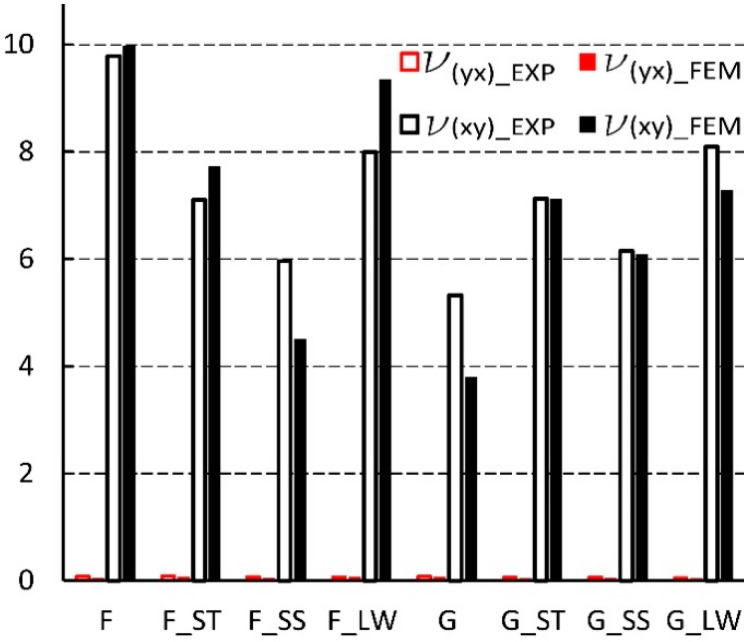
Poisson’s ratio for cells of the type: F, G (EXP-experimental research, FEM-numerical calculations).

**Figure 15 materials-13-02058-f015:**
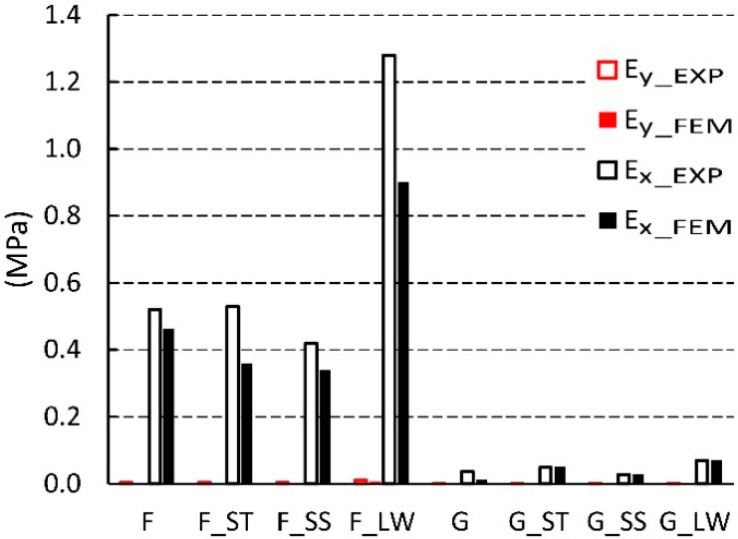
Linear modulus of elasticity for F, G cells (EXP-experimental research, FEM-numerical calculations).

**Figure 16 materials-13-02058-f016:**
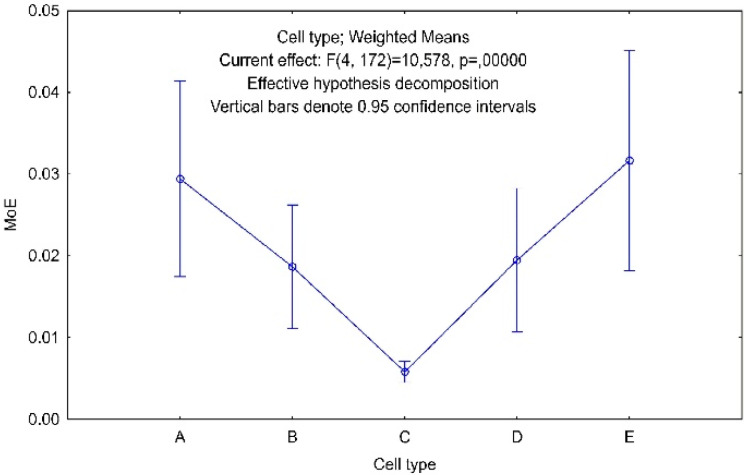
Tukey HSD test result for factor Cell type.

**Figure 17 materials-13-02058-f017:**
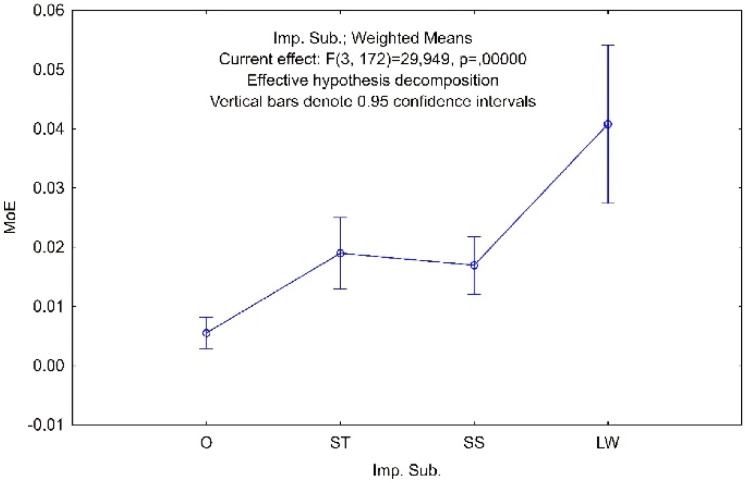
Tukey HSD test result for factor Imp. Sub.

**Figure 18 materials-13-02058-f018:**
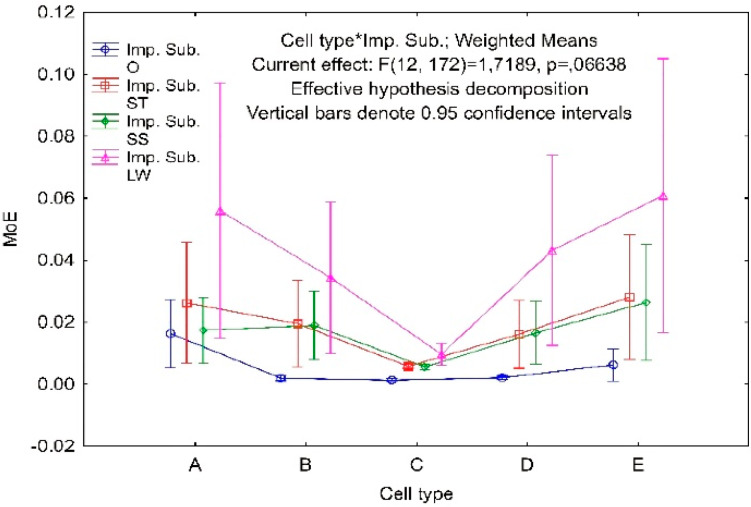
The combined effect of the type of cell and the method of impregnation on the modulus of elasticity.

**Table 1 materials-13-02058-t001:** Grammage of paper used.

Paper Thickness (mm) Agent	0.1	0.15	0.25
	Paper Grammage (g/m^2^)		
Test zero-without agent	85	123	134
Acetylated starch (ST)	90	139	178
Sodium silicate (SS)	101	150	207
LiquidWood^®^ (LW)	148	178	256

**Table 2 materials-13-02058-t002:** Dimensions of cells used for research, where: *ρ*—cell relative density, *S_y_*—cell width, *L_x_*—cell length, *l*—cell wall length, *h*—common cell wall length, *t*—cell wall (paper) thickness, *φ*—cell wall angle, *ε*—angle of the bisector of the angle between the cell walls (Figure 1a).

Cell Type	*ρ*	*S_y_*	*L_x_*	*I*	*h*	*t*	*φ*	*ε*
	(–)	(mm)					(˚)	
A	0.0249	16.62	23.99	11.5	3.9	0.15	45	67.5
B	0.0249	17.91	25.99	10.2	8.0	0.15	30	60.0
C	0.0249	23.28	20.15	12.2	6.0	0.15	20	55.0
D	0.0249	17.08	28.89	9.7	9.7	0.15	30	60.0
E	0.0249	13.33	46.48	13.0	12.0	0.15	60	75.0
F	0.0585	9.47	46.84	6.3	19.1	0.25	45	67.5
G	0.0344	6.21	33.74	6.0	11.7	0.10	60	75.0

**Table 3 materials-13-02058-t003:** Elastic properties of cores determined experimentally (EXP) (SD–standard deviation).

Cell Type	$\upsilon_{yx}/\mathrm{SD}$	$\upsilon_{xy}/\mathrm{SD}$	$E_{y}/\mathrm{SD}$	$E_{x}/\mathrm{SD}$
	(–)		(MPa)	
A	0.210/0.030	3.01/0.64	0.0025/0.0005	0.0274/0.0084
A_(ST)	0.250/0.060	2.52/0.33	0.0057/0.0013	0.0519/0.0117
A_(SS)	0.310/0.040	2.33/0.14	0.0035/0.0002	0.0313/0.0025
A_(LW)	0.090/0.010	3.95/0.25	0.0023/0.0007	0.1097/0.0174
B	0.320/0.030	1.77/0.25	0.0009/0.0001	0.0029/0.0009
B_(ST)	0.330/0.050	1.52/0.17	0.0043/0.0007	0.0385/0.0034
B_(SS)	0.420/0.050	1.80/0.06	0.0046/0.0003	0.0333/0.0045
B_(LW)	0.160/0.020	2.42/0.23	0.0025/0.0005	0.0661/0.0111
C	0.560/0.030	1.03/0.12	0.0008/0.0001	0.0019/0.0002
C_(ST)	0.800/0.010	0.93/0.07	0.0050/0.0006	0.0065/0.0004
C_(SS)	0.720/0.370	0.77/0.07	0.0053/0.0015	0.0059/0.0013
C_(LW)	0.550/0.050	1.28/0.14	0.0049/0.0007	0.0144/0.0004
D	0.360/0.030	1.94/0.31	0.0014/0.0001	0.0028/0.0002
D_(ST)	0.250/0.050	2.20/0.49	0.0026/0.0009	0.0296/0.0089
D_(SS)	0.320/0.030	1.86/0.28	0.0040/0.0007	0.0291/0.0770
D_(LW)	0.180/0.010	3.49/0.69	0.0040/0.0003	0.0824/0.0173
E	0.060/0.010	8.72/0.50	0.0006/0.0001	0.0119/0.0023
E_(ST)	0.070/0.010	7.24/1.12	0.0018/0.0002	0.0544/0.0070
E_(SS)	0.110/0.010	6.22/1.14	0.0017/0.0003	0.0510/0.0055
E_(LW)	0.270/0.010	8.12/0.21	0.0032/0.0007	0.1185/0.0163
F	0.074/0.021	9.93/1.13	0.0034/0.0012	0.5200/0.1900
F_(ST)	0.091/0.017	7.10/1.12	0.0044/0.0009	0.5300/0.1000
F_(SS)	0.067/0.008	5.96/0.37	0.0030/0.0007	0.4200/0.0700
F_(LW)	0.060/0.010	7.99/0.81	0.0108/0.0015	1.2800/0.0500
G	0.074/0.009	5.32/0.40	0.0007/0.0001	0.0371/0.0049
G_(ST)	0.053/0.009	7.16/0.93	0.0007/0.0001	0.0487/0.0056
G_(SS)	0.063/0.010	6.15/1.29	0.0006/0.0001	0.0272/0.0033
G_(LW)	0.051/0.015	8.09/1.19	0.0007/0.0001	0.0689/0.0084

**Table 4 materials-13-02058-t004:** Comparison of real (R) and theoretical (T) cell dimensions.

Cell Type	A		B		C		D		E		F		G	
Test	R	T	R	T	R	T	R	T	R	T	R	T	R	T
*l* (mm)	10.68	11.50	9.93	10.20	11.67	12.20	9.41	9.70	12.98	13.00	6.52	8.00	5.13	6.00
*h* (mm)	4.70	3.90	8.32	8.00	6.58	6.00	9.86	9.70	11.35	12.00	16.30	17.00	12.50	11.70
*φ* (°)	59.19	45.00	45.58	30.00	32.18	20.00	44.24	30.00	74.23	60.00	57.47	45.00	58.60	60.00

**Table 5 materials-13-02058-t005:** Univariate Tests of Significance–ANOVA.

Effect	ss	df	ms	f	p
Intercept	0.082457	1	0.082457	234.4467	0.000000
Cell type	0.014881	4	0.003720	10.5777	0.000000
Imp. sub.	0.031600	3	0.010533	29.9487	0.000000
Dir.	0.062411	1	0.062411	177.4510	0.000000
Cell type*Imp. sub.	0.007255	12	0.000605	1.7189	0.066376
Error	0.060494	172	0.000352		

ss—sum of squares, df—degrees of freedom, ms—mean sum of squares, f—test value, p—probability level.

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
