# Peer review of "Experimental Research and Numerical Analysis of the Elastic Properties of Paper Cell Cores before and after Impregnation"

_materials, 2020, doi:10.3390/ma13092058_

Round 1

Reviewer 1 Report

The topic of the specific experimental work is interesting, scientifically sound and provides new information and knowledge on the field of honeycomb panels. In my opinion, it could be published after some changes that will improve further the quality of the manuscript.

In key words, there is no need to use words included in the title. Additionally, instead of the "experiment", which is very general, you could add stiffness, rigidity, cell geometry or other useful words.

You should make a check in the whole text of the manuscript for grammatical and syntactical errors, such as line 29, 31, 53, 60, 75, 100, 104 etc. In line 43, you should check the meaning, since it is not clear ("the shrinking resources of natural fibres"). I believe that in some points, the authors could improve the writing to be more appropriate/scientific (please make a check in introduction chapter). In line 63, probably you mean indicating or proving? (the meaning is not very clear).

The introduction chapter, as the other chapters, is well prepared and provides the reader with the necessary information. However, describing the state of the art and referring some of the previous studies, you provided only some generic information, about the tests results of the boards made with the core paper of different impregnation techniques, substances etc.  

I would like also to comment on the opportunities of recycling and the environmental friendliness of the boards using the specific impregnation substances,that you mention in the materials and methods chapter, that I have no doubts on those, but what about PVAc adhesive? 

The results are well presented, justified, the statistical analysis seems to be complete and the conclusions chapter highlights the most significant points of the experiment. 

Author Response

Dear Editor and the Reviewers,

First of all, we want to thank for offering the authors the opportunity to respond to the reviewers comments. Below, step by step, we present our explanations and changes introduced to the text. All changes (in manuscript) are marked in red.

Responses to Reviewer #1:

We are grateful to the Reviewer for careful reading the manuscript and helpful remarks. Below we respond to the remarks one by one.

1. In key words, there is no need to use words included in the title. Additionally, instead of the "experiment", which is very general, you could add stiffness, rigidity, cell geometry or other useful words.

A: The authors modified the keyword list.

2. You should make a check in the whole text of the manuscript for grammatical and syntactical errors, such as line 29, 31, 53, 60, 75, 100, 104 etc. In line 43, you should check the meaning, since it is not clear ("the shrinking resources of natural fibres"). I believe that in some points, the authors could improve the writing to be more appropriate/scientific (please make a check in introduction chapter). In line 63, probably you mean indicating or proving? (the meaning is not very clear).

A: The authors checked on the format of the English language. In lines 43 and  63, the sentences were improved.

3. The introduction chapter, as the other chapters, is well prepared and provides the reader with the necessary information. However, describing the state of the art and referring some of the previous studies, you provided only some generic information, about the tests results of the boards made with the core paper of different impregnation techniques, substances etc.

A: The authors ones again checked the available literature. According to Reviewer #1 suggestion, we improved this part of the manuscript by a little bit much extensive explanation of the research results.

4. I would like also to comment on the opportunities of recycling and the environmental friendliness of the boards using the specific impregnation substances,that you mention in the materials and methods chapter, that I have no doubts on those, but what about PVAc adhesive?

A: PVAc Woodmax FF12.47 class D2 adhesive is a white glue which is an emulsion of polyvinyl acetate in water and is biodegradable. PVAc is non-toxic to humans. However, it does release toxic fumes if burned. These sentences were added to the text.

Jerzy Smardzewski

Reviewer 2 Report

The topic and results are generally interesting. There are only minor points, that should be corrected to improve the manuscript. I miss the discussion with results of similar studies. I believe that authors are able to find them and compare. The analysis of research results with three (54,56,57) sources and self-citations is not sufficient.

There are several English and formal changes that should be addressed through the paper:

L13: I prefer to use the term "epoxy resin" instead of LiquidWood. Parameters of used commercial epoxy resin (LiqiudWood) should be more explained in Materials and Methods.

L60: boards

L152-155: these sentences do not belong here, they should be re-styled and located above; within the description the main purpose.

L369, 372, 373, 374, etc. decimal dot

L614 use the term epoxy resin instead of liquid wood 

Author Response

Dear Editor and the Reviewers,

First of all, we want to thank for offering the authors the opportunity to respond to the reviewers comments. Below, step by step, we present our explanations and changes introduced to the text. All changes (in manuscript) are marked in red.

Responses to Reviewer #2:

We are grateful to the Reviewer for careful reading the manuscript and helpful remarks. Below we respond to the remarks one by one.

1. The topic and results are generally interesting. There are only minor points, that should be corrected to improve the manuscript. I miss the discussion with results of similar studies. I believe that authors are able to find them and compare. The analysis of research results with three (54,56,57) sources, and self-citations is not sufficient.

A: The reviewer's comment is correct. It would be beneficial for the work to refer to the results of other authors' rams. Admittedly, there are many works on the properties of impregnated paper. However, the results of this work do not apply to honeycomb cells. Therefore, several works from the last decade of the 21st century were selected, in which the use of phenolic resin impregnated honeycomb paper was described:

“It is difficult for authors to compare the obtained results with the other researches. No papers were found describing the impact of the paper core cell impregnation method on its elastic properties. In work [58] dynamic testing was conducted on phenolic GFRP and phenolic-impregnated aramid paper honeycomb specimens in order to analyse the effect of loading rate on the mechanical behavior. The strain rate effect for the composite material is remarkable and leads to increasing Young’s modulus, tensile strength, tensile failure strain, shear modulus and shear strength. Zinno et al. [59] explained that E-glass-phenolic composite shows a considerable reduction in ultimate strain and stress for the hot-wet and acid conditionings, but the Nomex honeycomb core shows a reduction in ultimate shear strength of about 20% for hygrothermal, freeze-thaw and acid conditionings. These works revealed better properties of impregnated cores. However, it is difficult to transfer the results obtained directly to the issue discussed in this paper”.

2. L13: I prefer to use the term "epoxy resin" instead of LiquidWood. Parameters of used commercial epoxy resin (LiqiudWood) should be more explained in Materials and Methods.

L60: boards

A: Epoxy resin (LiquidWood®) is a product protected by Abatron company. Therefore, the chemical composition of this substance is unknown. From this reason, the text was supplemented only with information that could be obtained on the manufacturer's website.

3. There are several English and formal changes that should be addressed through the paper:

L152-155: these sentences do not belong here, they should be re-styled and located above; within the description, the main purpose.

L369, 372, 373, 374, etc. decimal dot

L614 use the term epoxy resin instead of liquid wood

A: The authors checked on the format of the English language.

Jerzy Smardzewski
